# Applications of Mechanochemically Prepared Layered Double Hydroxides as Adsorbents and Catalysts: A Mini-Review

**DOI:** 10.3390/nano9010080

**Published:** 2019-01-08

**Authors:** Jun Qu, Lin Sha, Chenjie Wu, Qiwu Zhang

**Affiliations:** 1Key Laboratory of Resources Green Conversion and Utilization of the State Ethnic Affairs Commission & Ministry of Education, College of Resources and Environmental Science, South-Central University for Nationalities, Wuhan 430074, China; qujun2018@scuec.edu.cn (J.Q.); shalin1994@163.com (L.S.); cjscun@mail.scuec.edu.cn (C.W.); 2School of Resources and Environment Engineering, Wuhan University of Technology, Wuhan 430070, China

**Keywords:** layered double hydroxides, mechanochemistry, application, review

## Abstract

Mechanochemically prepared layered double hydroxide (M-LDH), which usually possesses high surface chemical activity and a substantial amount of surface defects, has presented outstanding application performance especially in the area of environmental protection. Recently published works on the mechanochemical synthesis of LDH were first introduced to provide a comprehensive summary on the preparation of the materials. Ensuing discussion provided an overview of recent research on the applications of M-LDH products as adsorbents and catalysts. The excellent adsorption performance and fast adsorption rate of the precursor of LDH produced by dry milling of raw materials was identified. The catalytic performances of M-LDH as catalysts, mainly photocatalysts, were then introduced. It is foreseeable that by rational utilization of mechanochemical processes and the unique chemical properties of M-LDH, increasing numbers of applications using M-LDH could be expected.

## 1. Introduction

The first identified layered double hydroxide (LDH) was Mg-Al LDH, which was found by Hochstetter in Sweden [1], with a chemical formula of Mg_6_Al_2_(OH)_16_CO_3_·4H_2_O. Allmann [2] confirmed the layered structure of LDH by testing the single-crystal structure in 1969. Due to the development of synthetic methods, the researchers have replaced Mg^2+^ with bivalent Fe^2+^, Cu^2+^, Zn^2+^, etc. and Al^3+^ with trivalent Fe^3+^, La^3+^, and even tetravalent Ti^4+^, Sn^4+^, etc. [3]. All the related compounds possessed a typical layered structure and were defined as a single kind of layered group minerals, which can be represented by a general formula, ${\left[M_{1-x}^{2+}M_{x}^{3+}{\left(\mathrm{OH}\right)}_{2}\right]}^{x+}{\left(A^{n}\right)}_{x/n}•{\mathrm{mH}}_{2}O$. LDH was also known as anionic clay for it was the only identified clay with an anion exchange property [4]. The interlayer anions of LDH (CO_3_^2−^, Cl^−^, SO_4_^2−^) can be exchanged by other inorganic and organic anions (such as CrO_4_^2−^, AsO_4_^3−^ [5], dodecyl sulfate [6], etc. [7]) in water. The researchers could tailor the composition of the layered sheet and adjust the species of the intercalated anions of LDH. The composition variability of LDH endowed its wide applications in various fields.

LDH can rarely be found in natural rocks. Two main synthetic routes have been developed to obtain the LDH product: (1) The liquid phase approach, which is the dominant method (e.g., co-precipitation [8] and hydrothermal synthesis [9]); (2) the solid-state approach—mechanochemical process [3]. However, high consumption of water and soluble heavy metal salts of the liquid-phase processes have led to the production of large amounts of polluted water. Additional treatment expenses have limited the production of LDH using the liquid phase approaches. On the contrary, LDH products manufactured by mechanochemical processes at the solid state have the advantage of no polluted water emissions. In 2007, Tongamp et al. [10] reported a mechanochemical approach to synthesize LDH, which was the first published data on a solvent-free process (dry and wet milling). Various kinds of LDH, such as Li-Al [11] LDH, Zn-Al [12] LDH, and Ca-Sn [13] LDH, etc., have been successfully prepared. Inorganic (tetraborate [14]) and organic matters (methyl orange [15], amino acid [13], and nonsteroidal anti-inflammatory [16]) have also been intercalated into the layered space of LDH by ball milling. The above work has demonstrated the flexibility of mechanochemistry to adjust the layered sheet component of LDH and the interlayer anions in the gallery space of LDH. Figure 1 displays the number of published papers on the topics of “mechanochemical”, “mechanochemical”, and “layered double hydroxide”, respectively. The number of published papers on mechanochemistry has grown rapidly in the past decades, which indicates that more and more researchers have been attracted to this scientific area. Originating from the work of Tongamp et al. [10], in 2007, the number of papers on the topics of “mechanochemical” and “layered double hydroxide” has showed a steadily increasing trend by each year. Although the total number of papers in this emerging field of research is relatively small now, it is foreseeable that more and more interesting phenomenon, new solid chemistry theories, various applications, etc. are to be discovered.

Different processes (e.g., ball milling, co-precipitation, hydrothermal) produce LDH with different basic properties (specific surface area, morphology, pore size distribution), which directly determined its application performance [17,18,19,20]. In the liquid phase, the metal ions could co-precipitate, which resulted in grain growth without interference. Therefore, the LDH product prepared in solution usually possesses a complete crystal structure, and a regular and uniform micro morphology [21]. The mechanochemical methods to synthesize LDH are based on applying high energy ball milling to conduct a solid reaction. The severe impact between balls and raw materials produces lattice distortion and dislocations, which results in a solid reaction between raw materials to form LDH [3]. On the macro level, the particles of LDH prepared by mechanochemistry usually agglomerate severely with irregular morphology. However, in microcosm, the M-LDH possesses substantial lattice distortion and structure faults, such as dislocations, which are crucial characteristics of catalysts [22]. Moreover, the ball milling process has been proved to be an effective way for surface doping [23], modification [24], manufacturing composite [25], etc. Compared with that of the liquid phase prepared LDH, a better application performance can be expected by rational utilization of the features of mechanochemical synthesis approaches and the unique properties of the M-LDH.

The application performance of LDH (manufactured by liquid phase approaches) for anionic pollutants’ removal (anions and oxyanions of the halogen elements [7], boron species [26], oxyanions [27], etc.), photocatalyst [28], and organic synthesis catalyst [17] have been extensively reviewed. However, no work has yet paid attention to reviewing the application performance of M-LDH as adsorbents and catalysts. In 2015, we conducted a review work on LDH synthesis by mechanochemistry, including different mechanochemical approaches to produce LDH and the intercalation of LDH [3,29,30]. At that time, the synthetic work on LDH was focused on Mg-Al LDH to explore the feasibility of different types of solid-state processes to produce LDH. Recently, the variety of mechanochemically prepared LDH has been greatly expanded. An introduction of the recently published works on the mechanochemical synthesis of LDH was first put forward to present comprehensive information on the synthesis of this material.

In this paper, the authors aim to:(1)Introduce the recently published works on the mechanochemical synthesis of LDH.(2)Review the application performance of LDH prepared by mechanochemistry as adsorbents and catalysts.(3)Discuss reasonable directions for future work.

## 2. LDH Synthesized by Mechanochemistry

### 2.1. One-Step Milling Process

Qu et al. [3] has reviewed the work on the one-step milling process to produce LDH. This one-step milling process has the advantage of easy operation, but the disadvantage of a low crystallinity of the product. Fahami et al. [31] solved the low crystallinity problem by aging the ground sample in the oven at 80 °C. MgCl_2_, AlCl_3_, Na_2_SO_4_, and NaOH were dry milled, washed, and aged in sequence. The characterization results of the obtained powder confirmed a highly crystalline phase of LDH with SO_4_^2−^ intercalated in the layered space [31]. A similar process was applied to produce Mg_0.8_Al_0.2_(OH)_2_Cl_0.2_ hydrotalcite using Mg(OH)_2_, AlCl_3_, and NaOH as raw materials. The formation and structural feature of Mg-Al-Cl LDH was controlled by the milling time. Five hour milling was optimal for the synthesis of LDH without any trace of brucite impurity [32]. Hexagonal-shaped hydrocalumite was manufactured by dry milling of CaCl_2_, AlCl_3_, and NaOH, followed by washing and aging operation. The interlayer space of the products was significantly influenced by the milling time. Sixty minutes of milling produced LDH with a relatively higher crystallinity with a dimension of 100–200 nm [33]. Fahami et al. [34] compared the features of Mg-Al-Cl LDH prepared by one-step milling with that of the hydrothermal process. The results demonstrated that the hydrothermal method could produce LDH with a higher crystallinity degree compared with that of the milled one. However, the mechanochemistry method was facile and simple, and thus could be applied to produce Mg-Al-Cl-LDH with an acceptable crystalline phase.

### 2.2. Two-Step (Dry and Wet Milling) Process

Tongamp et al. [10] introduced a two-step grinding operation (as illustrated in Figure 2) to synthesize Mg-Al, in which the Mg(OH)_2_ and Al(OH)_3_ was first activated into the amorphous state by dry milling for one hour and then the amorphous matter was wetly ground for two hours with water (just the amount of the crystal water) to achieve the crystallization of LDH. When replacing the water in the second step with salt (containing crystal water), such as Mg(NO_3_)_2_∙6H_2_O, a specific anion pillared LDH could be manufactured [35]. One-step wet milling of Mg(OH)_2_, Al(OH)_3_, and the needed water resulted in severe adhesion phenomenon (the sample strongly adhered to the bottom of the pot and the surface of balls). This meant that the added water more or less existed freely in the sample, not in the crystal structure of LDH, bonding the powder together. A considerable amount of unreacted raw materials could be observed by samples prepared through the one-step wet milling process, which was confirmed by X-ray diffraction (XRD) characterization. In the two-step milling process, a dry milling operation produced an amorphous state precursor, which could directly react with water in the second wet-milling process to provide a high crystalline LDH phase as a fine powder product. The published data by Tongamp proved that the dry milling operation was essential for effective production of LDH [10].

Utilizing the two-step dry and wet milling process, Qu et al. [11] synthesized Li-Al-OH in which LiOH and Al(OH)_3_ was first dry ground and then wet milled with the exact needed crystal water. Unlike the traditional solvent “imbibition” process to synthesize Li-Al LDH, which required excessive and high concentrated Li^+^, the mechanochemical process only needed the exact amount of the required molar ratio of the raw materials (according to the chemical formula of Li-Al LDH) to manufacture LDH. A fine powder product with no impurity phase of Al(OH)_3_ or LiOH was manufactured after the wet milling operation. Kuramoto et al. [36] applied the two-step milling process to manufacture acetate intercalated Mg-Al LDH. The brucite and gibbsite were first activated by dry milling and then magnesium acetate tetrahydrate was added to the mixture for further grinding. The obtained sample could create a stable aqueous suspension, which was used to produce thin films with varied thickness by casting on the substrates.

Ferencz et al. [37] attempted to manufacture Ca-Al LDH by manual grinding and ball milling of Al(OH)_3_ and Ca(OH)_2_. The manual milling operation failed to produce Ca-Al LDH, but the two-step milling process gave a pure phase of Ca-Al LDH. However, characterization data of the LDH product demonstrated that a carbonate (CO_3_^2−^) type of LDH was obtained. This meant carbon dioxide adsorption occurred during the milling operation. It has been proven that grinding Ca(OH)_2_ with Al(OH)_3_ can easily produce katoite (Ca_3_Al_2_(OH)_12_), instead of Ca-Al LDH [38,39]. The adsorption of CO_2_ was the key to fabricating Ca-Al LDH. To illustrate the mechanochemical synthesis mechanism of Ca-Al LDH, Qu et al. [40] studied the effect of anion addition on the synthesis of Ca-Al LDH. Figure 3 displays the XRD patterns of Ca-Al-CO_3_ LDH prepared by a two-step milling operation with different Ca(OH)_2_/CaCO_3_/Al(OH)_3_ molar ratios. Milling Ca(OH)_2_ with Al(OH)_3_ produced cubic katoite, Ca_3_Al_2_(OH)_12_, instead of LDH. When a third phase (CaCO_3_ or CaCl_2_) was added and ground together with Al(OH)_3_ and Ca(OH)_2_, the cubic katoite was transformed into Ca-Al-CO_3_ and Ca-Al-Cl LDH, respectively.

Milling Mg(OH)_2_ or LiOH with Al(OH)_3_ easily produced hydroxyl type Mg-Al-OH [10] and Li-Al-OH LDH [11]. However, cubic katoite rather than hydroxyl type LDH was achieved when Al(OH)_3_ was ground with Ca(OH)_2_ [40]. The hydration process of C3A (3CaO•Al_2_O_3_) fast-hardening cement hydration, as shown in Figure 4, could be used to explain the solid-state reaction pathways during ball milling for the synthesis of Ca-Al LDH.

Figure 4 illustrates that the hydration process of C3A would result in different phases under different environments. C3A with H_2_O alone first formed the metastable hydroxyl type Ca-Al-OH LDH, which would quickly transform to cubic katoite with a temperature over 30 °C. When other anions (Cl^−^, SO_4_^2−^, CO_3_^2−^) took part in the hydration process, LDH was obtained at the end rather than katoite. Corresponding to the hydration process of C3A, the reaction pathway for the mechanochemical synthesis of Ca-Al can be expressed as Equations (1) and (2).

Involving no other component:(1)$$3\mathrm{Ca}{\left(\mathrm{OH}\right)}_{2}+2\mathrm{Al}{\left(\mathrm{OH}\right)}_{3}\to {\mathrm{Ca}}_{3}{\mathrm{Al}}_{2}{\left(\mathrm{OH}\right)}_{12}$$

The third phase addition:(2)$$3\mathrm{Ca}{\left(\mathrm{OH}\right)}_{2}+2\mathrm{Al}{\left(\mathrm{OH}\right)}_{3}+{\mathrm{CaCl}}_{2}+{\mathrm{xH}}_{2}O\to {2\mathrm{Ca}}_{2}\mathrm{Al}{\left(\mathrm{OH}\right)}_{6}\mathrm{Cl}\cdot {\mathrm{xH}}_{2}O$$

Ferencz et al. [13] conducted a two-step milling craft (dry and wet milling) to synthesize Ca-Sn LDH using Ca(OH)_2_ and SnCl_4_∙6H_2_O as raw materials. Because of the large precipitation pH value difference between Ca^2+^ and Sn^4+^, the traditional co-precipitation process failed to manufacture Ca-Sn LDH. This proved the feasibility of mechanochemistry to overcome the difficulties for ions’ co-precipitation and produced a new binary LDH product. Although the authors did not explain the solid reaction pathways during the milling operation, it could be concluded that Sn(OH)_4_ and CaCl_2_ were firstly produced by the reaction between Ca(OH)_2_ and SnCl_4_∙6H_2_O, and then Ca(OH)_2_, Sn(OH)_4_, and CaCl_2_ were induced to react with each other, forming Ca-Sn LDH. We believe that the newly formed CaCl_2_ played the role of the third phase addition for the stabilization of the LDH phase. A similar process has been applied for the synthesis of Ca-Fe LDH using Ca(OH)_2_ and FeCl_3_∙6H_2_O as raw materials [41]. Ca(OH)_2_ firstly reacted with FeCl_3_∙6H_2_O to produce the phase of Fe(OH)_3_ and CaCl_2_, and then the residual Ca(OH)_2_, the newly formed Fe(OH)_3_, and CaCl_2_ together formed the Ca-Fe LDH. The newly formed CaCl_2_ was the key to fabricating Ca-Fe LDH by ball milling. This concept of involving a third phase to stabilize the desired compound may be applied to the synthesis of other entirely new LDH and also the adsorption of specific anions’ pollutants (taking the pollutants as the needed third phase).

### 2.3. Two-Step (Dry Milling and Agitation in Water) Process

The two-step milling process (dry milling and agitation in water) has displayed great potential as an alternative solvent-free method to replace the traditional co-precipitation methods for the synthesis of different LDH products. However, due to the severe impact of ball milling, the LDH product prepared by mechanochemistry always depicts an irregular morphology as shown in Figure 5. In the second wet milling process, the unreacted residual water made the powder aggregate together.

To gain a high crystalline LDH product with regular shapes, Qu et al. [12,15] replaced the second wet milling operation with agitation in water to prepare Zn-Al and Cu-Al LDH. An illustration of the dry milling and agitation in water process is displayed in Figure 6.

Zinc carbonate hydroxide hydrate (Zn_4_CO_3_(OH)_6_·H_2_O) was first dry ground with Al(OH)_3_ to obtain an amorphous precursor, which possessed a low crystallinity phase of LDH, indicating the formation of the basic structure of LDH. The precursor was then agitated in water for non-interfering crystallization to form high crystalline Zn-Al LDH. This process used carbonate and hydroxide as raw materials, which were much cheaper and more stable than that of the soluble salts used in the traditional co-precipitation methods. The same procedure was applied for the synthesis of Cu-Al LDH using basic cupric carbonate and Al(OH)_3_ as raw materials and therefore high crystalline Cu-Al LDH could be prepared [15]. The water in the second agitation step was replaced by methyl orange (MO) solution to manufacture an organic molecule intercalated LDH. The precursor prepared by dry milling could effectively adsorb MO into the gallery space of LDH, forming a pillared LDH product. Under the same conditions, the mechanochemical method could intercalate much more organic matters into the LDH structure than that of the traditional ion-exchange method [15]. The unique property of the precursor was vital for an outstanding adsorption performance, which will be reviewed in the next section.

Wang et al. [42] conducted a work to synthesize Mg-Al LDH using Mg(OH)_2_ and Al(OH)_3_ as the raw materials by the milling-agitation process. It was found that a considerable amount of unreacted Mg(OH)_2_ was observed in the product when directly milling Mg(OH)_2_ with Al(OH)_3_. Pre-milling of Mg(OH)_2_ was essential to transform stable Mg(OH)_2_ to an activated state for the complete solid reaction with Al(OH)_3_. The published data confirmed that the raw materials should be simultaneously transformed to an amorphous state during dry milling to realize a complete solid reaction for the synthesis of a pure phase LDH.

### 2.4. Two-Step (Dry Milling and Ultrasonic Treatment) Process

To gain a high crystalline LDH, the group of István Pálinkó combined the milling operation with ultrasonic treatment. Szabados et al. [43] worked on the ultrasonically-enhanced mechanochemical synthesis of CaAl-layered double hydroxides. The Ca(OH)_2_ and Al(OH)_3_ were first dry ground to an activated state and then irradiated by ultrasonic treatment in water, which contained appropriate anions, such as CO_3_^2−^, F^−^, Cl^−^, etc. The product showed high crystalline Ca-Al LDH intercalated by different inorganic anions. The same process has been applied for the synthesis of Zn-Al LDH [44]. ZnCl_2_ was precipitated by NaOH solution to gain Zn(OH)_2_-ZnO, which was then milled with Al(OH)_3_ to achieve the precursor of Zn-Al LDH. After ultrasonic irradiation, the precursor was transformed to high crystalline Zn-Al, with an unusual rose-like morphology. The molar ratio of Zn/Al in the product could be either 1/2 or 1/1 by controlling the initial Zn/Al ratio of the raw materials.

To sum up, dry milling of raw materials is essential to effectively prepare LDH. The second step can be either wet-milling, agitation in the water operation, or ultrasonic treatment. It can be concluded that the two-step mechanochemical process can not only realize the controllable adjustment of the components of the LDH sheet, but also intercalate different inorganic or organic anions into the layered space of LDH for the synthesis of different kinds of LDH products.

## 3. Application Performance of Mechanochemically Prepared LDH

### 3.1. Absorbent

It is well known that mechanochemical activation operation increases the specific area and the number of surface defects of the materials. Such changes contribute to the significant enhancement of adsorption performance [45,46]. A similar phenomenon has been observed on M-LDH. Guo and Reardon [47] prepared meixnerite by two-step grinding of MgO and Al(OH)_3_ for the removal of F^−^. After the adsorption, nordstrandite and sellaite were detected in the samples by XRD characterization. The results illustrated that the obtained samples could adsorb anions by an ion-exchange mechanism and also form a new phase with the anions on the surface of the adsorbent due to its outstanding surface activity.

Based on the newfound phenomenon from the mechanochemical synthesis of LDH, two aspects of work evaluated the adsorption performance of M-LDH. One was the utilization of the LDH precursor (possessing the basic structure of LDH) prepared by dry milling for anions’ removal; the other was a flexible use of the phenomenon of the third phase addition to synthesizing Ca-Al LDH for anionic pollutants’ adsorption (taking the pollutants as the needed third phase).

#### 3.1.1. Adsorption Performance of the LDH Precursor

Dry milling of the raw materials produced the amorphous precursor, which could easily produce LDH in water at room temperature, which meant that the precursor possessed the basic structure of LDH. The dry milling operation induced the solid reaction between raw materials to form the basic chemical bonding of LDH, which could be directly used as an anionic adsorbent for polluted water purification.

Wang et al. [42] pre-milled Mg(OH)_2_ to an activated state, which was then co-ground with Al(OH)_3_ to manufacture the precursor of Mg-Al LDH. The obtained sample showed an excellent adsorption capacity (1110.2 mg/g) towards methyl orange (MO), as shown in Figure 7. Dispersing the precursor in the polluted water at room temperature realized in situ adsorption of the pollutant together with the formation of LDH, with a significant increase of the removal efficiency. The XRD pattern of the sample (after MO adsorption) depicted a pure phase of Mg-Al LDH without the intercalated one, indicating a surface adsorption mechanism for the removal of MO. The precursor could also be applied for the purification of benzene homologue polluted water. He et al. [48] realized phenols’ removal from water by the precursor of Mg-Al LDH, with an excellent adsorption capacity of 82.6 mg/g phenols and 356.4 mg/g p-nitrophenol, through a mechanism of surface adsorption. The data demonstrated that the disorderly precursor could provide more active sites than other adsorbents, with a stable crystalline structure for the adsorption of anionic pollutants.

The hydroxyl type LDH could not pillar organic molecules into the gallery space of LDH. However, the precursor of carbonate type LDH prepared by dry milling showed both surface adsorption and an intercalation mechanism for the removal of organic matters. Dry milling malachite and gibbsite with a Cu/Al molar ratio of 2/1 produced the precursor of Cu-Al LDH. High adsorption activities of the precursor of Cu-Al LDH toward MO and dodecyl sulfate anion (DS^−^) were observed [15,49]. Through utilization of this unique property of the Cu-Al LDH precursor, Qu et al. [15,49] manufactured a pure phase of MO and DS^−^ intercalated Cu-Al LDH. The traditional ion-exchange process was also done for comparison. The quantitative elements’ analysis of the products proved that the precursor could uptake much more organic molecules into the structure of LDH than that of the well-formed LDH synthesized by the ion-exchange process. The results illustrated the outstanding performance of the precursor for loading organic molecules.

The ball milling process can produce nanoscale primary particles, which, however, usually agglomerate into micron-sized agglomerates. This restrains the adsorption capacity of the precursor. To scatter the severe agglomeration of mechanochemically prepared adsorbent, Ai et al. [50] introduced stable ZnO and SiO_2_ into the mechanochemical synthesis of Zn-Al LDH precursor. Promotions of removal efficiencies of 16.46% and 19.26% for methyl orange adsorption were achieved when ZnO (10% addition) and SiO_2_ (25% addition) were, respectively, added. The data manifested that the additive depressed the agglomeration phenomenon in the ball milling process and therefore promoted the adsorption efficiency of the precursor on methyl orange.

Besides the above application on pollutants’ adsorption, the precursor of LDH together with H_3_PO_4_ modified kaolinite could also be used for desalination of difficult alkali metals salts. The precursor of Mg-Al LDH, prepared by the milling of Mg and Al hydroxides, was used to incorporate nitrate anions through an exchange with the interlayer OH^−^, while the H_3_PO_4_ modified kaolinite exchanged K^+^ by H^+^ from the solution. The simultaneous use of the cationic (modified kaolinite) and anionic (LDH precursor) adsorbents realized a synergistic effect to increase the removal efficiency of potassium nitrate. A neutralization reaction took place in the suspension, allowing the continuous ions’ exchange for effective potassium nitrate fixation. The obtained sample after adsorption could serve as a potash and nitrogen compound fertilizer for agriculture [51].

The precursor prepared by dry milling can be used for pollutants’ removal, the fabrication of organic-inorganic composites [14,15,52], as well as selective adsorption for ion separation [53]. It is foreseeable that the dry-milling prepared LDH precursor will exhibit various excellent properties in different fields.

#### 3.1.2. Pollutants Adsorption as the Needed Third Phase

In Section 2.2, it was illustrated that a third phase addition transforms the katoite to LDH. This phenomenon was utilized by Zhong et al. [54] for the removal of Cr(VI) from aqueous solution. The adsorption mechanism is shown in Figure 8. Dry milling of Ca(OH)_2_ with Al(OH)_3_ produced a phase of cubic katoite. Dispersing the ground sample in CrO_4_^2−^ solution resulted in the Ca-Al-CrO_4_^2−^ LDH. Unlike the traditional LDH product with an ion-exchange mechanism for pollutants’ adsorption, the milled sample could directly react with the anions in the solution, forming the layered structure of LDH. A faster adsorption rate and higher capacity than that of the traditional adsorbent was achieved by the ground sample.

Improving the selectivity of the adsorbent towards target pollutants is important for practical application. Because of competitive adsorption, adsorption performance of adsorbents is extensively depressed. However, the data from Zhong et al. [54] showed that the coexistence of chloride (Cl^−^) in the solution, an element that is widely found in the natural environment, slightly weakened the adsorption performance of the Ca-Al-X precursor toward CrO_4_^2−^. The Ca-Al-X precursor presented excellent adsorption selectivity toward Cr (VI).

Carbonate and chloride is easily adsorbed into the layered structure of LDH while bromide and iodide are hard-to-intercalating. Szabados et al. [43] irradiated the precursor prepared by dry milling of Ca(OH)_2_ and Al(OH)_3_ with ultrasonic treatment in a solution containing CO_3_^2−^, F^−^, Cl^−^, Br^−^, and I^−^. The specific anions were intercalated into the layered sheet of Ca-Al LDH. Similarly, Szabados et al. [55] dry milled FeO(OH) (prepared by precipitation of FeCl_3_·6H_2_O with 25 wt% aqueous NH_3_ solution) with Ca(OH)_2_ to obtain the precursor sample in the first step. The precursor was then ultrasonically irradiated in water containing sodium salts (NaF, NaCl, NaBr, NaI, NaNO_3_, NaClO_4_, NaN_3_, and Na_2_CO_3_). The ultrasonically-enhanced milling craft successfully prepared the inorganic anions’ intercalated Ca-Fe LDH. The salts in the solution acted as the third phase for the LDH synthesis and were in-situ removed by the precursor. The above works by Szabados et al. present a promising way for the purification of halogen contaminated water, especially for water with radioactive I (131) contamination.

### 3.2. Photocatalyst

The excellent anionic organics’ affinity of the precursor could not only be used to produce intercalated LDH, but also adsorb and photocatalytic decolorize anionic dye. Qu et al. [56] prepared a Zn-Al LDH precursor photocatalyst by dry milling Zn_4_CO_3_(OH)_6_·H_2_O and Al(OH)_3_ for the adsorption and photocatalytic decoloration of MO. The whole process is shown in Figure 9. The precursor of Zn-Al LDH easily intercalated the MO molecules into the interlayer space of LDH, resulting in a higher adsorption capacity than that of the well-formed LDH. The adsorption behavior of the precursor also promoted the photocatalytic performance toward MO decoloration. The contact between the organic matters and the photocatalyst was essential for effective photocatalytic degradation. The intercalated MO molecules could be degraded by the photocatalytic active sites of the inner space of the precursor. In other words, the precursor provided more photocatalytic active sites than that of the well-formed LDH, which resulted in the higher photocatalytic degradation efficiency of the precursor than that of the LDH.

Dispersing other photocatalysts on the LDH matrix is an effective way for the enhancement of photocatalytic degradation efficiency [57]. The LDH matrix, with excellent adsorption property, enriches the organic pollutants into the structure of the LDH, which is then degraded by the supported photocatalyst under light irradiation. Li et al. [58] ground the Mg(OH)_2_, Al(OH)_3_, CdCl_2_, and Na_2_S together, followed by stirring in water. Mg(OH)_2_ reacted with Al(OH)_3_, forming the Mg-Al LDH matrix. CdCl_2_ and Na_2_S formed the semiconductor, CdS, with a visible light response property. The size of the synthesized CdS particles was nanoscale and they were uniformly distributed on the surface of Mg-Al LDH. Compared with the Mg-Al LDH or nanostructured CdS, the composite exhibited better photocatalytic performance under visible light irradiation for the degradation of (Methylene blue) MB.

Composites of different materials can produce a photocatalyst with an outstanding photocatalytic performance [59]. The LDH ball milling process was proved to be an effective way for manufacturing composites [60]. Li et al. [61] reported a one-step mechanochemical craft to produce Ag/Zn-Al LDH with excellent photocatalytic performance. Zinc carbonate hydroxide hydrate, Al(OH)_3_, and elemental silver was directly dry ground and then agitated in water. The characterization results confirmed the uniform distribution of Ag nanoparticles on the surface of Zn-Al LDH. The obtained photocatalyst formed a Schottky barrier between Ag and LDH under visible light illumination, producing hydroxyl radical and superoxide radical to degrade methyl orange. Through utilization of this facile and environmental friendly ball milling method, Li et al. [62] produced nanosized Zn*_x_*Cd_1−*x*_S/Zn-Al layered double hydroxide heterojunctions by grinding zinc carbonate hydroxide hydrate, CdCl_2_, Al(OH)_3_, and Na_2_S and then agitating it in water. The obtained composite with a heterostructure effectively restrained the recombination of photoelectrons and holes, which contributed to the enhancement of the photocatalytic activity for the degradation of MO compared with the pure Zn-Al LDH precursor or Zn*_x_*Cd_1−*x*_S under the same conditions.

### 3.3. Other Applications

Tongamp et al. [63] manufactured Ni-doped LDH for the generation of hydrogen gas from polyethylene (PE) by a two-step milling operation of Mg(OH)_2_, Ni(OH)_2_, and Al(OH)_3_. The obtained LDH was then ground with PE and heated under the temperatures of 450 and 550 °C. This generated 49.4% (mass yield) of gaseous compounds of H_2_, CH_4_, CO, and CO_2_ with a concentration of 80.6%, 15.2%, 3.0%, and 1.3%, respectively. This simple operation of milling could be utilized to transform hydrocarbons-contained solid waste into gas fuel. Pavel et al. [64] oxidized olefin by the lanthanum-doped Mg-Al LDH, which was separately synthesized by co-precipitation and mechanochemistry. The experimental data confirmed a better catalytic performance of the co-precipitation sample than that of the mechanochemical one.

The weak oxygenated intermediates’ adsorption ability of NiFe hydroxide (NiFe-LDH) limited its catalytic activity for efficient water oxidation catalysis. A ball milling method for the generation of tensile strain was conducted by Zhou et al. [65] to enhance the binding strength of NiFe hydroxide to oxygenated intermediates. High energy ball milling of NiFe-LDH introduced the tensile strain to the structure, which upshifted the d band center and made their anti-bonding states less filled. This change improved the OER activity, including an earlier OER onset and faster current growth.

The above work demonstrates the potential application of M-LDH for energy and organic synthesis. However, the usage of M-LDH is mainly focused around environmental protection, as shown in Section 3.1 and Section 3.2. Only a few works in other fields were published. More work needs to be done to expand the application area (such as water splitting and organic synthesis) of M-LDH, and an outstanding application performance could be expected because of the unique properties of M-LDH (high surface chemical activity and a considerable amount of surface defects).

## 4. Towards Future Work

As a newly arising field, applications of mechanochemical methods to synthesize LDH and its application performance as an adsorbent and catalyst were reviewed. Dry milling of raw materials produces the precursor of LDH, which possesses the basic structure of LDH, with strong adsorption activity. The phenomenon of the third phase addition to inducing the solid reaction of LDH formation was confirmed. Based on the newfound sights from the mechanochemical synthesis, the excellent adsorption performance and extremely fast adsorption rate of the precursor were identified. By rational utilization of the feature of mechanochemical synthesis approaches and the unique properties of the LDH prepared by ball milling, a photocatalyst with high catalytic activity could be produced.

As summarized above, more and more reports related to the mechanochemical synthesis of LDH are available. Through the utilization of the unique properties of the M-LDH (high surface chemical activity, surface defects), its outstanding performance in different fields has been gradually identified. However, performance evaluation work remains restricted to the area of environmental protection. By rational utilization of the typical chemical properties of the milling prepared LDH, more application fields, such as water splitting and organic synthesis, with outstanding performance can be expected.

## Figures and Tables

**Figure 1 nanomaterials-09-00080-f001:**
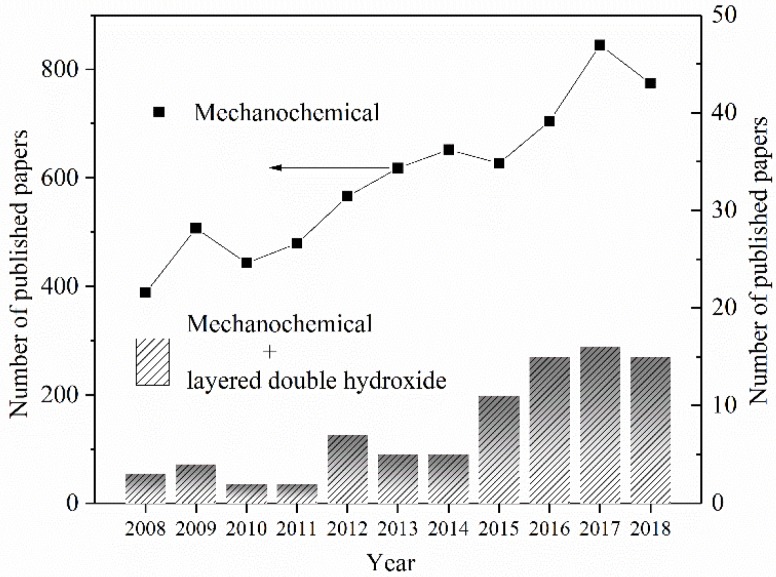
The number of published papers with topics on mechanochemical, mechanochemical, and layered double hydroxide, respectively (literature search in the database of Web of Science).

**Figure 2 nanomaterials-09-00080-f002:**
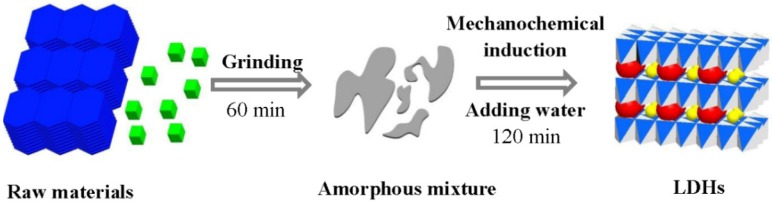
Illustration of the two-step grinding operation. Reproduced with permission from [3]. Copyright Elsevier, 2016.

**Figure 3 nanomaterials-09-00080-f003:**
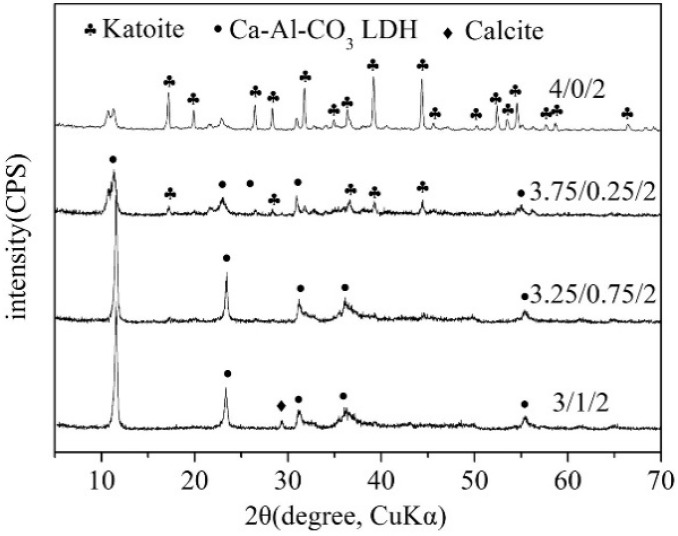
X-ray diffraction (XRD) patterns of Ca-Al-CO_3_ layered double hydroxide (LDH) prepared by a two-step milling operation with different Ca(OH)_2_/CaCO_3_/Al(OH)_3_ molar ratios. Reproduced with permission from [40]. Copyright Elsevier, 2016.

**Figure 4 nanomaterials-09-00080-f004:**
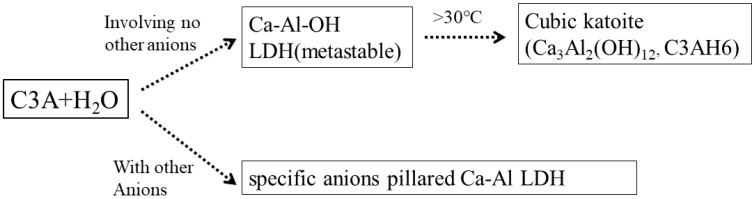
The hydration reaction pathways of C3A (3CaO•Al_2_O_3_) fast-hardening cement.

**Figure 5 nanomaterials-09-00080-f005:**
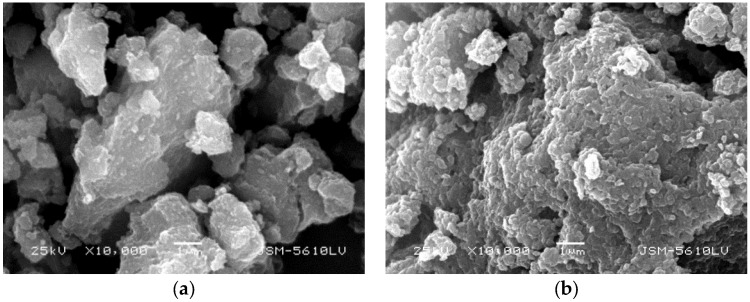
Scanning electron microscope (SEM) images of Li-Al LDH (**a**) and Ca-Al LDH (**b**) prepared by the two-step grinding approach.

**Figure 6 nanomaterials-09-00080-f006:**
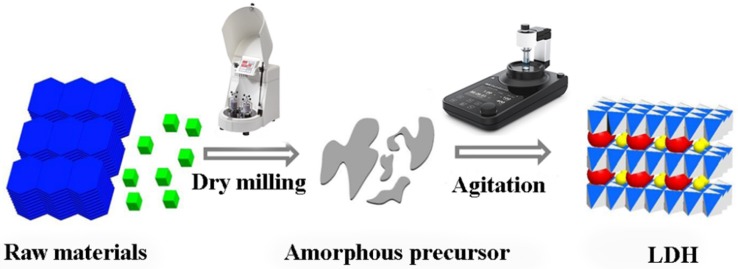
Illustration of two-step dry grinding and agitation in the water process operation. Reproduced with permission from [12]. Copyright Elsevier, 2017.

**Figure 7 nanomaterials-09-00080-f007:**
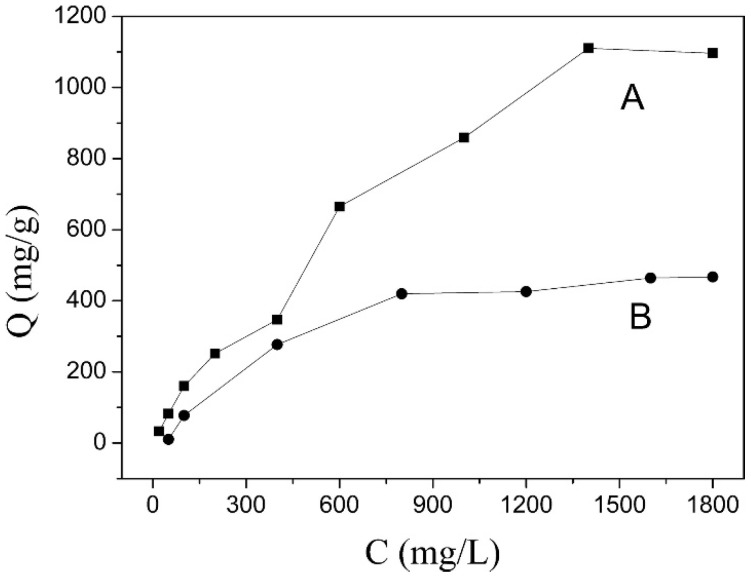
Adsorption isotherms of methyl orange (MO) on the precursor (**A**) and LDH (**B**). Reproduced with permission from [42]. Copyright Elsevier, 2016.

**Figure 8 nanomaterials-09-00080-f008:**
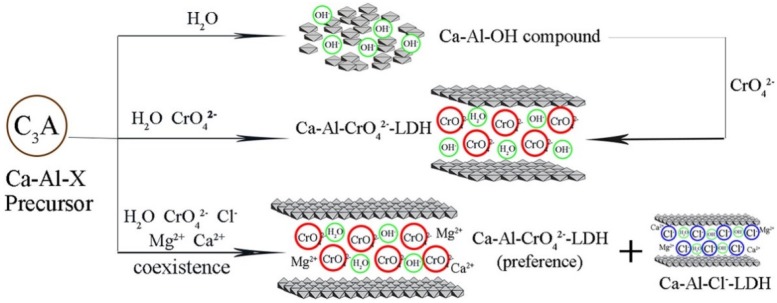
Cr(VI) adsorption mechanism by the precursor of Ca-Al LDH. Reproduced with permission from [54]. Copyright Elsevier, 2017.

**Figure 9 nanomaterials-09-00080-f009:**
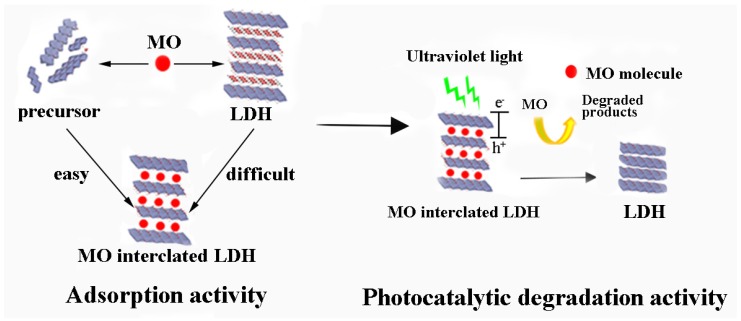
Precursor of Zn-Al LDH for enhanced adsorption and photocatalytic decoloration of MO [56].

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
