# Peer review of "Applications of Mechanochemically Prepared Layered Double Hydroxides as Adsorbents and Catalysts: A Mini-Review"

_nanomaterials, 2019, doi:10.3390/nano9010080_

Reviewer 1 Report

The present mini review article prepared by Zhang et. al. aims at discussing the recent developments in mechanochemmical synthesis of LDH, their adsorption performance and finally discuss potential future directions.

I am afraid that as it is written, the review falls a bit short in terms of the excepted information the reader was hoping to get. In general the review is badly written which makes most of its parts unreadable. In general the manuscript suffers from multiple grammatical and linguistic errors.

A few points for the authors who might want to consider:

1.   It is supposed that the abstract sets the scene for what it will follow. This abstract does not say anything. The authors should provide the reader the reasons and the limits/content of the review article.

2.   There are numerous confusing sentences: eg. "Taking the adsorption capacity of LDH as an example, the sample prepared by hydrothermal method usually possessed higher BET surface area and smaller particle size than that of the product manufactured by co-precipitation that usually results in a better adsorption performance of the hydrothermal sample [12,13]." Too long sentence and I am not sure what the authors are trying to say here.

3.   Page 3, line 117: "... the final steady phase of calcium aluminum compound..." Not sure what the auhtors are trying to say here.

4.   The authors use very often the word "craft" which I can't figure out in what context is being used. Please use appropriate terms. In that aspect, the subheadings in 2.2 and  2.3 are both "two step crafts"? and in parentheses some description. The content within the parentheses gives useful information to the reader.

5.   Page 6, line 211: "...To excavate the adsorption ability of ball milling produced LDH..." I don’t have any idea what the authors are trying to say here.

6.   Page 6, line 213: "...other one was a flexible use of the phenomenon of the third phase to synthesize Ca-Al LDH for anionic pollutants adsorption." What is the thirs phase phenomenon? I barely understand again this sentence

7.   The authors need to give the description of the abbreviations used in the manuscript eg. Page 6, line 223: MO--> ?

8.    Page 7, line 259: Anions as third phase? The presence of anions does not constitute a different phase

9.    Page 7, line 270: "...the coexistence of high concentration Cl- slightly weakened the fixation of CrO42- indicating its excellent selectivity toward Cr(VI) and also a great potentiality for practical application." Again this sentence is very confusing and I am not sure what is the message the authors are trying to convey.

10.  Page 8, Conclusions/perspectives: "...the phenomenon of the third base..." What is the phenomenon the authors are talking about?

11.  Page 8, Conclusions/perspectives: "The unique properties of the LDH prepared  by ball milling for the application have been gradually excavated." I don’t understand the meaning of this sentence. 

12.  Page 8, Conclusions/perspectives: The conclusions fail providing to the reader the important take home messages. What exactly the reader learns from this review article? Also, the authors do not give what is the future work that can be done and what other directions remained to be explored.

Overall, this review article seems like a half-baked effort. The authors need to give to the authors the background information hidden in the papers they discussed. Also, they authors fail to discuss how the meachnochemical synthetic approach influence the adsorption properties of the LDH materials. they need to provide the necessary connections.

I am afraid I cant recommend publication of the manuscript at this stage. The review article might appropriate for publication only after major revision.

Author Response

Thank you so much for kindly reviewing our manuscript and giving meaningful comment. We deeply appreciate the time and effort to point out the English usage problems. We have carefully checked every sentence in the manuscript and revised the manuscript according to your comments as follows:

Q1. It is supposed that the abstract sets the scene for what it will follow. This abstract does not say anything. The authors should provide the reader the reasons and the limits/content of the review article.

To provide the readers a comprehensive information of our reviewing work, we have rewritten the abstract as Mechanochemically prepared layered double hydroxide (M-LDH), which usually possessed high surface chemical activity and a considerable amount of surface defects, has presented outstanding application performance especially in the area of environmental protection. This work provided an overview of recent research on the application of the M-LDH products as adsorbent and catalyst. To present a comprehensive information on the materials preparation, the recently published works on the mechanochemical synthesis of LDH were first introduced. Ensuing discussions reviewed the excellent adsorption performance and extremely fast adsorption rate of the precursor of LDH which was produced by dry milling of raw materials. The mechanochemical methods for the preparation of the catalyst, mainly photocatalyst, and its photocatalytic performance were introduced. It is foreseeable that more application field with an outstanding performance of M-LDH could be expected by rational utilization of the feature of mechanochemical crafts and the unique chemical properties of M-LDH.”

Q2. There are numerous confusing sentences: eg. "Taking the adsorption capacity of LDH as an example, the sample prepared by hydrothermal method usually possessed higher BET surface area and smaller particle size than that of the product manufactured by co-precipitation that usually results in a better adsorption performance of the hydrothermal sample [12,13]." Too long sentence and I am not sure what the authors are trying to say here.

Thanks a lot to point out our English writing problem. We have broken the very long sentences in the manuscript for an easy understanding. The changes are listed as follows.

In line 53-59  “Li-Al[15] LDH, Zn-Al[16] LDH, Ca-Sn [17] LDH, et al have been successfully prepared by a similar two-step milling process, and various inorganic (tetraborate [18]) and organic matters (methyl orange [19], amino acid [17] and nonsteroidal anti-inflammatory [20]) have been intercalated into the layered space of LDH by ball milling, that proved the ability of mechanochemistry to adjust not only the layered sheet component of LDH but also the interlayer anions in the gallery space of LDH.” for “Various kinds of LDH, such as Li-Al[11] LDH, Zn-Al[12] LDH, Ca-Sn [13] LDH, et al have been successfully prepared. Inorganic (tetraborate [14]) and organic matters (methyl orange [15], amino acid [13] and nonsteroidal anti-inflammatory [16]) have also been intercalated into the layered space of LDH by ball milling. The above work proved the ability of mechanochemistry to adjust the layered sheet component of LDH and also the interlayer anions in the gallery space of LDH.”

In line 88-95 The sentence of “Taking the adsorption capacity of LDH as an example, the sample prepared by hydrothermal method usually possessed higher BET surface area and smaller particle size than that of the product manufactured by co-precipitation that usually results in a better adsorption performance of the hydrothermal sample [12,13]." was deleted.

        In line 346-348 "In section 2.1,it has been illustrated that the third phase addition transformed the katoite to LDH phase which was utilized by Zhong et al. [42] for the removal of Cr(VI) from aqueous solution.” for “In section 2.2, it has been illustrated that the third phase addition transformed the katoite to LDH phase. This phenomenon was utilized by Zhong et al. [54] for the removal of Cr(VI) from aqueous solution.”

        In line 366-370 "Unlike the traditional LDH product with an ion-exchange mechanism for the adsorption process, the milled sample could react with the anions in the solution forming the layered structure of LDH which contributed to a faster adsorption rate than that of the traditional adsorbent.” for “Unlike the traditional LDH product with an ion-exchange mechanism for pollutants adsorption, the milled sample could directly react with the anions in the solution forming the layered structure of LDH. A faster adsorption rate and higher capacity than that of the traditional adsorbent could be achieved by the ground sample."

       In line 389-392 “Szabados et al. [43] firstly dry milled FeO(OH) (prepared by precipitation of FeCl3·6H2O with 25 wt% aqueous NH3 solution) with Ca(OH)2 to get the precursor sample which was then ultrasonically irradiated in water containing sodium salts (NaF, NaCl, NaBr, NaI, NaNO3, NaClO4, NaN3 and Na2CO3).” for “Szabados et al. [55] dry milled FeO(OH) (prepared by precipitation of FeCl3·6H2O with 25 wt% aqueous NH3 solution) with Ca(OH)2 to get the precursor sample in the first step. The precursor was then ultrasonically irradiated in water containing sodium salts (NaF, NaCl, NaBr, NaI, NaNO3, NaClO4, NaN3 and Na2CO3).”

Q3. Page 3, line 117: "... the final steady phase of calcium aluminum compound..." Not sure what the auhtors are trying to say here.

        Fig .3 was added to help explain the third phase addition for the synthesis of Ca-Al LDH. Milling Ca(OH)2 with Al(OH)3 produced cubic katoite, Ca3Al2(OH)12, not LDH. However, when the third phase of CaCO3 or CaCl2 was added, the LDH product could be obtained. In Fig.4, we applied the hydration process of C3A (3CaO•Al2O3, a fast-hardening cement) hydration to explain the solid-state reaction pathways during ball milling for the synthesis of Ca-Al LDH. The hydration process of C3A would result in different phases in the different environment. C3A with H2O involving no other components first formed the metastable hydroxyl type Ca-Al-OH LDH which would quickly transform to cubic katoite with the temperature over 30 °C. when other anions (Cl-, SO42-, CO32-) took part in the hydration process, there obtained the LDH phase at the end, not katoite. Relevant illustration has been added in line 192-202.

 Q4. The authors use very often the word "craft" which I can't figure out in what context is being used. Please use appropriate terms. In that aspect, the subheadings in 2.2 and 2.3 are both "two step crafts"? and in parentheses some description. The content within the parentheses gives useful information to the reader.

In line 53, 72, 202, 202, 231, 240, descriptions about the word “craft” have been added for the accurate expression.

Q5. Page 6, line 211: "...To excavate the adsorption ability of ball milling produced LDH..." I don’t have any idea what the authors are trying to say here.

In line 298-300, the sentence has been corrected as “Based on the newfound phenomenon form the mechanochemical synthesis of LDH, two aspects work have been done to evaluate the adsorption performance of M-LDH.”

Q6. Page 6, line 213: "...other one was a flexible use of the phenomenon of the third phase to synthesize Ca-Al LDH for anionic pollutants adsorption." What is the thirs phase phenomenon? I barely understand again this sentence

Milling Ca(OH)2 with Al(OH)3 produced cubic katoite, Ca3Al2(OH)12, not LDH. When the third phase (CaCO3 or CaCl2) was added and ground together with Al(OH)3 and Ca(OH)2, the cubic katoite was transformed into the Ca-Al-CO3 and Ca-Al-Cl LDH respectively.  The relevant illustration has been corrected as “the other one was a flexible use of the phenomenon of the third phase addition to synthesizing Ca-Al LDH for anionic pollutants adsorption (taking the pollutants as the needed third phase).” In line 302-305.

Q7. The authors need to give the description of the abbreviations used in the manuscript eg. Page 6, line 223: MO--> ?

       In line 246, abbreviating methyl orange as MO was put forward. MO was then used in the following text representing methyl orange.

Q8. Page 7, line 259: Anions as third phase? The presence of anions does not constitute a different phase.

       The phase of “Anions adsorption as the needed third phase” was corrected as “Pollutant Adsorption as the needed third phase”

Q9.  Page 7, line 270: "...the coexistence of high concentration Cl- slightly weakened the fixation of CrO42- indicating its excellent selectivity toward Cr(VI) and also a great potentiality for practical application." Again this sentence is very confusing and I am not sure what is the message the authors are trying to convey.

       The expression has been corrected as “The data from Zhong et al [54] showed that the coexistence of chloride (Cl-) in the solution, an element that could be widely found in the natural environment, slightly weakened the adsorption performance of the Ca-Al-X precursor toward CrO42-. The Ca-Al-X precursor presented excellent adsorption selectivity toward Cr (VI). “in line 375-379.

Q10.  Page 8, Conclusions/perspectives: "...the phenomenon of the third base..." What is the phenomenon the authors are talking about?

Milling Ca(OH)2 with Al(OH)3 produced cubic katoite, Ca3Al2(OH)12, not LDH. When the third phase (CaCO3 or CaCl2) was added and ground together with Al(OH)3 and Ca(OH)2, the cubic katoite was transformed into the Ca-Al-CO3 and Ca-Al-Cl LDH respectively. Fig.3 was added to help to illustrate this phenomenon. The anionic pollutants in the solution could also act as the third phase for the formation of intercalated LDH as shown in section 3.1 (2)). By this mechanism, the pollutants could be removed.

Q11.  Page 8, Conclusions/perspectives: "The unique properties of the LDH prepared  by ball milling for the application have been gradually excavated." I don’t understand the meaning of this sentence. 

       The sentence was corrected as “Utilization the unique properties of the M-LDH (high surface chemical activity, surface defects), its outstanding performance has been gradually identified.” in line 472-474.

Q12.  Page 8, Conclusions/perspectives: The conclusions fail providing to the reader the important take home messages. What exactly the reader learns from this review article? Also, the authors do not give what is the future work that can be done and what other directions remained to be explored.

We agreed that the “Conclusion” was substandard and rewrote this part in line 461-477.

 Overall, this review article seems like a half-baked effort. The authors need to give to the authors the background information hidden in the papers they discussed. Also, they authors fail to discuss how the meachnochemical synthetic approach influence the adsorption properties of the LDH materials. they need to provide the necessary connections.

To give specific background information and the necessary connections between mechanochemistry and adsorption properties of LDH, more illustration has been given in the “Introduction” part:

In line 40-42: "The researchers could tailor the composition of the layered sheet and also adjust the species of the intercalated anions of LDH. The composition variability of LDH endowed itself wide applications in various fields.”

In line 47-51: "The former one played the dominant role in the synthesis of LDH. However, the massive water and soluble heavy metal salts consumption of the liquid-phase processes produced a large amount of polluted water. The high environmental costs limited the production of LDH by the liquid phase approaches. On the contrary, the mechanochemical processes manufactured LDH products at solid state without polluted water emission."

     In line 61-69: "Fig.1 displays the number of published papers with topics on “mechanochemical”, “mechanochemical” and “layered double hydroxide" respectively. The number of published papers about mechanochemistry grew rapidly in the last ten years indicating that more and more researchers have engaged in this area for science research and exploration. Originating from the work of Tongamp et al [10] in the year 2007, the number of papers with topics on “mechanochemical” and “layered double hydroxide” has been steadily increasing every year. Although the total number of papers in this emerging field of research was relatively small now, it is foreseeable that more and more interesting phenomenon, new solid chemistry theories, significant application, et al. will be dug out.

     In line 75-87: "In the liquid phase, the metal ions could co-precipitate and realize the grain growth without interference. That is to say, the LDH product prepared in the solution usually possessed complete crystal structure, regular and uniform micro morphology [21]. The mechanochemical methods to synthesize LDH usually applied high energy ball milling to conduct the solid reaction. The severe impact between balls and raw materials produced lattice distortion, dislocations which resulted in the solid reaction between raw materials to form LDH phase [3]. On the macro level, the particles of LDH prepared by mechanochemistry usually agglomerated severely with irregular morphology. But in microcosm, the M-LDH possessed considerable lattice distortion and structure fault such as dislocations which were crucial for catalysts [22]. Also, the ball milling process has been proved to be an effective way for surface doping [23], modification [24], manufacturing composite[25], et al. Comparing with that of the liquid phase prepared LDH, a better application performance could be expected by rational utilization of the feature of mechanochemical synthesis approaches and the unique properties of the M-LDH."

Reviewer 2 Report

Following points should be considered.

Typical milling time for dry milling and wet milling should be given in the Figure 1 and main text.

The reviewer cannot agree the expression at line 186 "The second step could be the wet-milling, agitation in water operation or ultrasonic treatment with the advantage of solvent-free,...".  Water is apparently solvent for reaction.

Advantages of the present process are clearly mentioned.  Mechanical milling process is the batch process, and it takes rather long time, and sample correction after milling is rather difficult.  Thus, the reviewer believes that this process could not be good for large scale (practical) production.

Author Response

Reviewer #2: Following points should be considered.

Thank you so much for kindly reviewing our manuscript and giving meaningful comment. We have revised the manuscript according to your comments as follows.

Typical milling time for dry milling and wet milling should be given in the Figure 1 and main text. The milling time has been added in Fig.1 and also in the main text at line 135.

The reviewer cannot agree the expression at line 186 "The second step could be the wet-milling, agitation in water operation or ultrasonic treatment with the advantage of solvent-free,...".  Water is apparently solvent for reaction.

The reviewer’s opinion was appropriate. The phase of “with the advantages of solvent-free, high crystalline and good dispersity product respectively” was deleted.

         Advantages of the present process are clearly mentioned.  Mechanical milling process is the batch process, and it takes rather long time, and sample correction after milling is rather difficult.  Thus, the reviewer believes that this process could not be good for large scale (practical) production. 

We agreed the viewpoint of the reviewer and the relevant expression on the practical production advantage of the mechanochemical process was deleted in line 155-156, 242-244.

Reviewer 3 Report

The manuscript « Mechanochemical synthesis of layered double hydroxide and its adsorption performance: a mini-review » nanomaterials-398049 by Q. Zhang et al. tries to gather information about firstly the mechanochemical synthesis of these materials and secondly their efficiency in the adsoprtion field.

Compared to the review article published in Applied Clay Sicence in 2016 (Applied Clay Science 119 (2016) 185-192 by the same authors in my point of view there is no real novelty justifying the publication of a new review article. The additional  part on adsorption performance mainly describes in a different light the publications detailled in the synthesis part.  Unfortunately, the authors do not provide a crtical point of view on the topic just describing the published results. Moreover, it is unfortunate that in such a small topic, the authors mainly focused on their personal publications omitting some other works on the topics (for instance the publications of Fahami et al).

In conclusion in my point of view, this review does not succeed to bring interesting scientific explantions about the effect of mechanosynthesis use on textural properties (surface are, particle size, aggregation state…) and the properties of adsorption.  The interest for the reader is then very poor.

Author Response

Reviewer #3: The manuscript « Mechanochemical synthesis of layered double hydroxide and its adsorption performance: a mini-review » nanomaterials-398049 by Q. Zhang et al. tries to gather information about firstly the mechanochemical synthesis of these materials and secondly their efficiency in the adsoprtion field.

Compared to the review article published in Applied Clay Sicence in 2016 (Applied Clay Science 119 (2016) 185-192 by the same authors in my point of view there is no real novelty justifying the publication of a new review article. The additional part on adsorption performance mainly describes in a different light the publications detailled in the synthesis part.  Unfortunately, the authors do not provide a crtical point of view on the topic just describing the published results. Moreover, it is unfortunate that in such a small topic, the authors mainly focused on their personal publications omitting some other works on the topics (for instance the publications of Fahami et al).

In conclusion in my point of view, this review does not succeed to bring interesting scientific explantions about the effect of mechanosynthesis use on textural properties (surface are, particle size, aggregation state…) and the properties of adsorption.  The interest for the reader is then very poor.

Thank you so much for kindly reviewing our manuscript and giving meaningful comment. We have revised the manuscript according to your comments.

       The review work (Applied Clay Science 119 (2016) 185-192) was conducted in the year 2015, At that time, the synthetic work on LDH was mainly about Mg-Al LDH which explored the feasibility of different types of solid-state processes to produce LDH phase. Recently, the variety of the mechanochemical prepared LDH has been greatly expanded. To present a comprehensive information on the synthesis of this material, an introduction of the recently published works on the mechanochemical synthesis of LDH were first put forward. The works from Fahami,Ferencz, Szabados et al. have also been included in the main text.

       To bring interesting scientific explanations about the effect of mechanochemical approaches, more information and illustration has been added to the “Introduction” part:

In line 40-42: "The researchers could tailor the composition of the layered sheet and also adjust the species of the intercalated anions of LDH. The composition variability of LDH endowed itself wide applications in various fields."

In line 47-51: "The former one played the dominant role in the synthesis of LDH. However, the massive water and soluble heavy metal salts consumption of the liquid-phase processes produced a large amount of polluted water. The high environmental costs limited the production of LDH by the liquid phase approaches. On the contrary, the mechanochemical processes manufactured LDH products at solid state without polluted water emission."

       In line 61-69: "Fig.1 displays the number of published papers with topics on "mechanochemical", "mechanochemical" and "layered double hydroxide" respectively. The number of published papers about mechanochemistry grew rapidly in the last ten years indicating that more and more researchers have engaged in this area for science research and exploration. Originating from the work of Tongamp et al [10] in the year 2007, the number of papers with topics on “mechanochemical” and “layered double hydroxide” has been steadily increasing every year. Although the total number of papers in this emerging field of research was relatively small now, it is foreseeable that more and more interesting phenomenon, new solid chemistry theories, significant application, et al. will be dug out."

       In line 75-87:"In the liquid phase, the metal ions could co-precipitate and realize the grain growth without interference. That is to say, the LDH product prepared in the solution usually possessed complete crystal structure, regular and uniform micro morphology [21]. The mechanochemical methods to synthesize LDH usually applied high energy ball milling to conduct the solid reaction. The severe impact between balls and raw materials produced lattice distortion, dislocations which resulted in the solid reaction between raw materials to form LDH phase [3]. On the macro level, the particles of LDH prepared by mechanochemistry usually agglomerated severely with irregular morphology. But in microcosm, the M-LDH possessed considerable lattice distortion and structure fault such as dislocations which were crucial for catalysts [22]. Also, the ball milling process has been proved to be an effective way for surface doping [23], modification [24], manufacturing composite[25], et al. Comparing with that of the liquid phase prepared LDH, a better application performance could be expected by rational utilization of the feature of mechanochemical synthesis approaches and the unique properties of the M-LDH."

      Besides reviewing the adsorption performance of mechanochemically prepared LDH, the photocatalytic properties, the catalysis for hydrogen production, organic synthesis and water oxidation were also summarized. We hoped that the newly added content could meet the requirement to be published in Nanomaterials.

Round  2

Reviewer 1 Report

The authors made a substantial effort to improve the overall quality of this review article.

There are still some minor points that need to be rectified:

The resolution of Figs 6 and 9 need to be improved.

Subheadings 2.3 and 2.4: Misuse of word craft again. plerase rephrase.

The manuscript needs to be edited for English once more.

I am happy to recommend publication after minor revision.

Author Response

Thank you so much for kindly reviewing our manuscript and giving positive comments. We sincerely appreciated your time and effort to help improve the quality of our manuscript.

  Fig.6 and Fig.9 have been redrawn to improve the resolution.

  The word “craft” in subheading 2.3 and 2.4 was replaced by “process”.

  For English writing, we have checked every sentence again for the accurate expression. And the corrections were listed as follows:

  In Line 1-4 “Application of Mechanochemically Prepared Layered Double Hydroxides as Adsorbents and Catalysts: a Mini-Review” for “Application of Mechanochemically Prepared Layered Double Hydroxide as Adsorbent and Catalyst: a Mini-Review”

  In line 22, “catalysts” for “catalyst”, “photocatalysts” for “photocatalyst”, “their” for “its”.

  In line 125, “the added water more or less” for “more or less added water”.

  In line 249-251, “As well known, mechanochemical activation operation could increase the specific area and the number of surface defects of the materials.” for “As well known, mechanochemical activation of materials could bring the changes of specific area increasing, more surface defects as adsorption site.”

Reviewer 2 Report

The reviewer believes that the reivsed manuscript can be accepted for publication, if this manuscript meets the scope of this journal.

Author Response

Thank you so much for kindly reviewing our manuscript and giving positive comments. We deeply appreciated your time and effort to help improve the quality of our manuscript.